# Cellular Senescence in Metabolic-Associated Kidney Disease: An Update

**DOI:** 10.3390/cells11213443

**Published:** 2022-10-31

**Authors:** Pan Gao, Xingjian Zou, Xin Sun, Chun Zhang

**Affiliations:** Department of Nephrology, Union Hospital, Tongji Medical College, Huazhong University of Science and Technology, Wuhan 430022, China

**Keywords:** cellular senescence, metabolic-associated kidney diseases, MetS, T2DM, hypertension, obesity, hyperuricemia

## Abstract

Cellular senescence is described as the state where the cell cycle is arrested irreversibly, which occurs in response to various forms of stress factors in cells, leading to the senescence-associated secretory phenotype (SASP). We can assess the accumulation of senescent cells in tissues or organs through biomarkers of cellular senescence such as p16^INK4a^, p53, p21, and SA-β-GAL. In recent decades, a large number of studies have reported the biomarkers of increased cell senescence in pathogenic tissues, demonstrating the possible connection between cell senescence and various diseases. Kidney damage often occurs in the pathophysiological process of certain metabolic diseases, resulting in metabolic-associated kidney diseases. For example, hypertension causes systemic arteriosclerosis, and the kidney can be seriously affected by abundant blood vessels, which may lead to a decreased glomerular filtration rate (GFR) and proteinuria, resulting in hypertension-related kidney diseases. The accumulation of senescent cells may also be observed in some metabolic-associated kidney diseases (such as obesity-related nephropathy, hypertension-related nephropathy, and diabetic nephropathy). In this paper, we review existing knowledge regarding the influence of cellular senescence on metabolic-associated kidney diseases, providing new ideas for future treatment.

## 1. Introduction

Cellular senescence is described as the irreversible state in which cell replication stops. Senescent cells not only quit the cell cycle, but also undergo many other phenotypic changes. The cumulative effect of these changes may finally be manifested as pathological changes in related tissues and organs that eventually lead to disease [1]. In this way, it can be divided into “acute senescence” and “chronic senescence”, where the former is observed in the process of steady-state damage and repair of tissues and organs. In various settings, acute senescence is generally a physiological, procedural process limiting renal fibrosis, improving immune surveillance and wound healing, and exerting positive effects on renal regenerative capacity [2]. The latter, “chronic senescence”, is the reaction that occurs in response to various forms of cellular stress or damage, including DNA damage or mutation, telomere shortening, high levels of reactive oxygen species (ROS) [3], mitochondrial dysfunction, epigenetic stress, inflammation, and so on. Although senescent cells withdraw from the cell cycle, they still maintain their metabolic function, releasing some cytokines and chemokines, and promoting the neighboring cell to enter the inflammatory state through paracrine signaling. As a result, the occurrence of cell senescence is scattered, which leads to the continuous accumulation of senescent cells. It is worth mentioning that chronic inflammation is an important feature of cellular senescence. Unlike acute inflammation, chronic inflammation can last a long time. Acute inflammation is caused by infection or injury, and it participates in the destruction of pathogens, repair of injury, and recovery of tissue function. Acute inflammation is usually well-controlled and disappears when balance is re-established in the body; however, in some diseases, factors such as the unsuccessful removal of harmful substances or failure of inflammatory regulation can lead to chronic inflammatory states. Tissues with cellular senescence are infiltrated by lymphocytes and macrophages, which will be induced by SASP to remove the senescent cells. In these tissues, pro-inflammatory cytokines (e.g., IL-1,2,6,8; tumor necrosis factor α, TNFα) and anti-inflammatory cytokines (e.g., interleukin-1 receptor antagonist, TGF-β) are secreted simultaneously. In the normal and healthy aging process, they are in a delicate balance; however, when this balance is broken, the cellular senescence process will accelerate and various senescence-related diseases can occur [4]. In the kidney, differing from acute senescence, under chronic senescence, the accumulation of senescent cells acts against recovery of the tissue and contributes to the chronic progression of the metabolic-associated kidney diseases mentioned above. In addition, it serves to reduce renal function—for example, through accelerating glomerulosclerosis, affecting the regenerative capacity of the kidney, and promoting renal allograft rejection [5]. The cellular senescence we discuss here refers to “chronic senescence”.

## 2. Cellular Senescence

### 2.1. Signaling Pathway of Cellular Senescence

There are two main pathways associated with cellular senescence: the p53/p21 and p16/Rb pathways [1]. When stress occurs (e.g., DNA damage), a series of cascade reactions are activated, contributing to the expression of response signals (ATM/ATR/ATF) to increase p53 expression, followed by the promotion of p21^CIP1^ (inhibitor of cell proliferation) expression. DNA damage can also increase the expression of p16^INK4a^. As cyclin-dependent kinases inhibitors (CDKIs), p21^CIP1^ and p16^INK4a^ inhibit CDK complexes (CDK2, CDK4/6), thus inhibiting the process by which Retinoblastoma protein (Rb) is phosphorylated into pRb by the CDK complex. Eventually, the release of the transcription factor E2F combined with Rb is obstructed, resulting in cell-cycle stagnation at the G1/S check point. Similarly, when chk1/2 is activated, the cell cycle will be blocked at the G2/M [2]. Based on this mechanism, some molecules can promote/inhibit cellular senescence through these pathways. Klotho, for example, is an anti-senescence protein that can inhibit senescence by inhibiting the p53/p21 pathway, making it a major regulator of cellular senescence. Another important regulatory factor is SIRT1, which is also an inhibitor of cellular senescence. It can inhibit the p53/p21 pathway through the deacetylation of p53 (Figure 1).

### 2.2. Senescence-Associated Secretory Phenotype (SASP)

Senescent cells usually secrete a series of molecules through autocrine/paracrine pathways, including pro-inflammatory factors (IL-1α, IL-1, IL-6, and IL-8), chemokines (CXCL-1/3, CXCL-10), growth factors (HGF, TGF-beta, GM-CSF), and proteases (metalloprotease). This group of molecules is called the Senescence-Associated Secretory Phenotype (SASP). On the one hand, SASP is a signal that requires the immune system of the body to clear out aging cells. They can induce lymphocytes, macrophages, and other cells to remove aging cells. On the other hand, they contribute to chronic low-grade inflammation and diseases. They can also act on the senescent cells to accelerate senescence, thus playing a functional role in senescence [3].

### 2.3. Detecting Cellular Senescence

The most common way to judge cell senescence is to detect biomarkers. From the pathways above we can obtain a direct, effective method: the detection of p53/p21^CIP1^ and p16^INK4a^. Results derived from some mouse models have supported the conclusion that p16^INK4a^-positive cells accumulate significantly in many tissues characterized by accelerated senescence. In addition to detecting CDKI, we can start with the consequence of cell senescence to detect SASP expression. However, the results may differ in different tissues, organs, and diseases. Therefore, it is essential to choose one or more suitable biomarkers [6]; for example, a commonly used SASP is TGF-β, which is a complex molecule exerting diverse effects. Combined with recent research, the relationship between TGF-β and cell senescence has been revealed gradually. TGF-β regulates the expression of downstream target genes, including many events related to cell aging, such as DNA damage or mutation, telomere shortening, and cell cycle regulation, among others. TGF-β signaling has been shown to have a significant impact on cellular senescence [4]. In addition, aging-related β-galactosidase (SA-β-GAL) is also an important biomarker for the detection of cellular senescence. It exists only in senescent cells, but not in resting cells, tumor cells, or immortalized cells. It can induce the activity of aging galactosidase at a pH of six.

## 3. Cellular Senescence in Metabolic-Associated Kidney Diseases

Metabolic syndrome (MetS), a complex of interrelated risk factors for cardiovascular disease and diabetes, is characterized by central obesity (increased waist circumference), hyperglycemia, dyslipidemia (high triglyceride blood levels and low high-density lipoprotein blood levels), and increased blood pressure. As an important metabolic organ, the kidney has a close relationship with metabolic syndrome. MetS usually aggravates kidney damage and causes or aggravates kidney pathologies, typically manifested as microalbuminuria and renal insufficiency. For example, severe obesity can lead to glomerular hypertrophy and glomerular sclerosis, leading to proteinuria. This is called obesity-related nephropathy, which was first discovered in 1974 by Weisinger. Diabetic capillary complications can lead to pathological changes to the kidney, thickening of the glomerular capillary basement membrane, and widening of the mesangium. Clinical manifestations can change from proteinuria to uremia. Similarly, hypertension-related nephropathy is a serious complication of hypertension, which is characterized by arteriosclerotic kidneys. Clinical manifestations include nocturia increasing, albuminuria, and finally, uremia. For hyperlipidemia, although there is no hyperlipidemia-related nephropathy, a large number of studies have shown that lipids have an effect on the proliferation and signal transduction of glomerular cells, and accelerate glomerulosclerosis through inflammatory reaction. These metabolic diseases often exist at the same time, and can promote each other; furthermore, they share a common pathophysiological basis: insulin resistance. It is widely believed that there exists a significant relationship between hyperglycemia, hypertension, and hyperlipemia.

The basic manifestations of metabolic syndrome are insulin resistance and the expansion of adipose tissue. Studies have shown that metabolic syndrome not only causes cardiovascular diseases, but also (directly or indirectly promotes) the development of chronic kidney diseases. Meanwhile, obesity, hypertension, and diabetes, which are closely related to metabolic syndrome, may also cause kidney damage. These results may eventually lead to secondary kidney diseases, such as obesity-related nephropathy, hypertension-related nephropathy, diabetic nephropathy, and hyperuricemia-related nephropathy. Cell senescence plays an important role in the development of metabolic nephropathy; for example, aging cells in the adipose tissue of obese patients have an impact on insulin resistance and impaired glucose tolerance, thus aggravating diabetes. Diabetes creates a microenvironment of hyperglycemia and promotes the senescence of endothelial progenitor cells (EPC) [7], thus aggravating vascular injury and atherosclerosis. Next, we discuss the role of cellular senescence in the development of metabolic-associated kidney diseases.

### 3.1. Obesity-Related Nephropathy

Obesity is now defined as a disease. According to WHO standards, obesity occurs at a BMI greater than 30. First, Veisinger reported on the relationship between severe obesity and proteinuria. In obesity, when proteinuria (possibly accompanied by microscopic hematuria) is detected, renal biopsy shows glomerular hypertrophy and focal glomerular sclerosis (FSGS)—while other diseases have been excluded—which can be considered obesity-related nephropathy. There is a connection between obesity and kidney disease. It has been estimated that 20–25% of kidney disease patients in the world are obese, and this number may be significantly increased after intermediate disease including type 2 diabetes mellitus (T2DM) and hypertension. Many epidemiological studies have shown that obesity is independently related to proteinuria, chronic kidney disease, Acute kidney injury (AKI), and End-Stage Renal Disease (ESRD) [5]. The results of a large-scale study have shown that, when the BMI increased to more than 25 kg/m, the risk of a decreased glomerular filtration rate was also increased. The specific mechanism through which obesity causes kidney damage and develops into obesity-related nephropathy has not yet been confirmed [6]. It is generally held that there are two ways by which it develops: direct and indirect, for example, some products secreted by adipocytes display nephrotoxicity. Obesity leads to an increase in intra-abdominal pressure, and the infiltration of fat into the kidney, which further leads to glomerular hypertension and activation of the renin–angiotensin–aldosterone system, thus promoting the occurrence of hypertension. On the other hand, obesity is an important risk factor for diabetes, and can indirectly cause kidney damage by promoting other disease states in MetS. These will be explained later in detail [8]. In these processes, cellular senescence also plays an important role in the disease. Obesity may speed up the induction of cell senescence while, at the same time, cellular senescence aggravates obesity. Both of these factors influence each other, and have a lot in common.

Obesity is usually due to excessive nutrient intake. It accelerates cell senescence through some internal reactions, which are part of the above-mentioned cell stress mechanism, such as telomere damage, oxidation, DNA damage, and so on, and which strongly promote cell senescence. In obesity, persistent low inflammation in adipose tissues is common, which is one of the characteristics of cell senescence. Senile preadipocytes release pro-inflammatory factors through paracrine and activate neighboring cells to a pro-inflammatory state. These senescent cells in adipose tissues may trigger the infiltration of immune cells, further aggravating inflammation. This vicious cycle leads to the accumulation of senescent cells [9]. On the other hand, cell senescence has an influence on obesity-related nephropathy. A major principle of MetS is the expansion of adipose tissues. One hypothesis is that excessive energy intake in MetS leads to the disorder of many physiological regulation systems, triggering inflammation and oxidative stress pathways [10], specifically, the expanded adipose tissue usually leads to increased free fatty acid (FFA) turnover [11,12]. In the case of insulin resistance, FFA mobilization (lipolysis) in adipose tissues is accelerated due to decreased insulin inhibition, leading to higher FFA in plasma [13,14,15]. Adipose tissue not only can store energy, but also plays an important endocrine function. It can regulate the dynamic balance of the whole body by regulating energy balance, insulin sensitivity, blood pressure, angiogenesis, and immune function. Obesity is related to the increase in adipose tissue. A large number of studies have shown that the aging of adipose tissue cells plays a positive role in insulin resistance and diabetic nephropathy [7]. Most overweight or obese people present insulin resistance [16], and, especially in obesity, a lack of physical activity and a diet leading to atherosclerosis are considered to be responsible for IR [17]. Senescent cells can secrete SASP, such as IL-6, TNF-α, and activin A, to promote insulin sensitivity. In obesity, due to the secretion of cytokines and chemokines (SASP) and the infiltration of macrophages, the inflammatory state can be observed in tissues. Tissue expansion is the main source of inflammatory cytokines. It has been shown that cell senescence partially determines the inflammation of adipose tissues [18].

A factor related to obesity and cell senescence is the tumor suppressor p53, which is the key regulator of adipogenesis. Under normal circumstances, it should be repressed before adipogenic precursor cells differentiate into insulin-reactive adipocytes. In obesity, p53 is activated and active oxygen is accumulated, interfering with normal adipogenic differentiation, potentially promoting the inflammatory reaction and insulin resistance [7].

There is a significant amount of research showing the connection between obesity and cellular senescence. For example, the research of Minminine et al. has reported that, in an obese mouse model with excessive calorie intake and T2DM, oxidative stress in adipose tissues was up-regulated, the expression of senescence markers such as p53 was increased, and SA-β-GAL and pro-inflammatory cytokines levels were increased. In addition, research has shown that p16^INK4a^, p53, and p21 were significantly increased in adipose tissue from mice fed a high-calorie diet for 4 months. In another study, DNA damage and cell senescence were artificially produced in adipose tissue in mice. The result indicated that the mice were obese and presented impaired glucose tolerance [7,19]. Using pifithrin-a to inhibit p53 resulted in reversal of the above changes. Allyson et al. have shown that the reduction in senescent cells or anti-aging treatment can improve the glucose tolerance of obese mice, enhance insulin sensitivity, promote fat generation, and reduce circulating inflammatory mediators [20].

Therefore, the above evidence proves the relationship between obesity-related nephropathy and cell senescence. Adipose tissue expansion caused by excessive nutrition will increase replication, enhancing various cell stress factors of cellular senescence (e.g., telomere damage, inflammation, and oxidative stress), and inducing the occurrence of cellular senescence in adipose tissues and cells. Senescent cells change the tissue structure through SASP, leading to insulin resistance and aseptic inflammation. The secreted SASP signals lead to cell senescence of the adjacent cells, thus promoting the occurrence of obesity, which, in turn, leads to clinical symptoms such as proteinuria, secondary to obesity nephropathy. In addition, the sterile inflammatory environment and insulin resistance, which are usually associated with obesity nephropathy, also contribute to the occurrence of diabetes. It is obvious that there is a close relationship between obesity and kidney diseases. Obesity can (directly or indirectly) induce obesity-related nephropathy. At present, the specific pathogenesis is not clear, but it appears that cell senescence is an important intermediate link in these processes, and more related mechanisms need to be determined.

### 3.2. Diabetic Nephropathy

Diabetic nephropathy is a microvascular complication of diabetes, in which the lesions are mainly located in the glomerulus. The GFR increases early, leading to a state of high filtration. With the development of the disease, edema, proteinuria, and hypertension appear gradually, eventually leading to renal insufficiency and even uremia.

Cellular senescence also plays an important role in diabetic nephropathy. Diabetes itself can create an environment that accelerates cellular senescence; in turn, cellular senescence will promote the progress of the disease. It is well-known that age and obesity are the main risk factors of T2DM. From the above, we know that obesity itself and cellular senescence in obesity can lead to insulin resistance, which is the key mechanism of T2DM. Pancreatic B cells are insulin-secreting cells. When insulin resistance occurs, the amount of insulin needed to maintain normal blood glucose will increases. T2DM occurs when the insulin secretion function of pancreatic B cells cannot meet the insulin demand. A large number of studies have shown that cellular senescence mainly inhibits the proliferation of B cells through the p16^INK4a^ pathway, leading to T2DM. Taschen’s research in 2009 has shown that p16^INK4a^ is regulated by the BMI-1 complex. B cells are up-regulated in mice with low p16^INK4a^ and high BMI-1. On the contrary, the proliferation of B cells is limited. In addition, the activity of CDK4 required for B cell proliferation was also inhibited by p16^INK4a^ through the above-mentioned pathway [21,22]. In contrast, in 2016, Helman et al. conducted an experiment to induce the stagnation of B cell proliferation by overexpressing p16^INK4a^. Their results proved that p16^INK4a^ can promote insulin secretion [23]. Some scholars have explained that simply increasing the single phenotype of increased p16^INK4a^ is insufficient to mimic cellular senescence. The stagnation of proliferation might cause B cells to compensate for insulin secretion by increasing the volume. More research is needed to validate the results of these experiments.

DN itself can also accelerate kidney aging. In DN, the accumulation of advanced glycation end-products (AGEs) may induce podocyte injury, glomerular mesangial cell apoptosis, and TGF-β expression, thus leading to renal insufficiency and accelerated aging [4]. AGEs stimulate the activation of AGEs receptors, which leads to oxidative stress and cell dysfunction, and stimulates the occurrence of cellular senescence. High glucose levels promote macrophage infiltration into the kidney, mitochondrial dysfunction, and activation of NOX1-PKC signaling, which leads to an increase in reactive oxygen species (ROS) and the aging of renal tubular epithelial cells and endothelial cells [7,9]. Hyperglycemia can also induce endoplasmic reticulum (ER) stress, leading to the senescence of renal tubular epithelial cells. Mesangial cells provide structural support for the glomerular capillary ring and regulate the ultrafiltration surface for glomerular filtration, which is directly affected by hyperglycemia. In cultured glomerular mesangial cells, a high extra-cellular glucose concentration promotes extracellular matrix (ECM) accumulation, cell cycle arrest, cell hypertrophy, and the induction of senescence [10]. Furthermore, studies have shown that, in a high-glucose environment, NOX activity and ROS production in mouse podocytes are increased, and cell apoptosis may occur [11]. Kidney cell senescence occurs, and, as a result, kidney-related clinical symptoms appear in DN. In addition, the clinical symptoms of DN include hypertension, as a result of renin–angiotensin–aldosterone system (RAAS) dysfunction. Abnormal activation of RAAS may lead to cellular senescence, as it can cause oxidative stress and/or the down-regulation of anti-aging proteins (e.g., Klotho and SIRT). Oxidative stress (OS) is also a key factor [12,13] that can lead to DN and increase senescence, and it plays an important role in MetS. In MetS, oxidative stress is characterized by the up-regulation of ROS, which are mainly derived from NADPH oxidase (NOX) 1,2,4. Due to metabolic factors, NOX leads to excessive ROS production in glomerular podocytes, endothelial cells, and mesangial cells, which may lead to the occurrence of cellular senescence and, eventually, can cause kidney lesions. In patients with DN, an increase in OS leads to a series of cellular events (e.g., DNA damage, endoplasmic reticulum stress), which will cause cellular senescence, as described above.

In summary, cellular senescence plays an important role in the pathogenesis of DN, which may be mediated by the inhibition of islet B cell proliferation through the p16^INK4a^ pathway. However, the determination of the specific mechanism still requires further study. In DN, there are many factors that can cause cell stress, thus accelerating the occurrence of cellular senescence.

### 3.3. Hypertension-Related Kidney Diseases

Hypertension-related nephropathy is one of the main complications of hypertension. Hypertension can cause systemic arteriosclerosis, and the kidney is seriously affected by nephron anatomy, which will lead to decreased GFR. Long-term hypertension can also cause dysfunction of glomerular capillary endothelial cells and injury of glomerular visceral epithelial cells, resulting in an increase in basement membrane permeability and proteinuria. The local and systemic inflammatory environment promoted by cell senescence promotes the development of cardiovascular disease. It is well-known that age is a major risk factor for hypertension and the occurrence of cellular senescence can be observed in almost all hypertensive patients or vascular tissues of experimental models. Cellular senescence is a common phenotype in hypertension. In hypertension, the decline of SASP and vascular function often occurs before actual aging. If cellular senescence is out of control, the relative age of blood vessels may increase, thus promoting the occurrence and development of hypertension. In pathological cardiovascular tissues, many studies have demonstrated an increase in aging markers, including p16 [22] and SA-β-GAL. Senescent cells may promote the progression of hypertension by mediating inflammation and oxidative stress; however, the specific mechanism needs to be verified through further studies. Some studies in the opposite direction can help us to speculate on the relationship between them. According to these studies, the removal of senescent cells can delay aging and restore organ function to some extent [24,25], including the vascular system and kidney [26]. Evidence has been presented that targeted aging therapies, such as drugs involving DNA damage repair mechanisms, cell cycle checkpoint kinases, and tumor suppressor factors, can reduce pathophysiology [27,28].

The article “Novel Contributors and Mechanisms of Cellular Aging in Hypertension-Associated Premature Vascular Aging” has integrated and reported a series of studies on aging in experimental models of hypertension into a table. Endothelial progenitor cells (EPCs) contribute to neovascularization [14,15,16,17], and the dysfunction of EPC is closely related to the development of hypertension. Studies have shown that, in hypertension, the function of EPCs is damaged [18], and some hypertension drugs can correct the functional damage of EPCs [19]. There are some studies on the mechanism leading to the dysfunction of EPCs in hypertension; however, the exact mechanism remains unclear. Haendeler et al. assessed the effect of antioxidant therapy, and their results showed that aging marker levels were decreased, proving the correlation between the accumulation of ROS and cellular senescence in endothelial cells [20]. Studies have proved that oxidized low-density lipoprotein and angII induce the cell senescence of EPCs through oxidative stress [21,22]. In addition, vascular oxidative stress has also been experimentally confirmed to induce hypertension [29,30,31]. Clinical studies have shown that the production of ROS is increased in patients with hypertension [24,32]. From this research, it appears EPCs are more likely to undergo cell senescence through oxidative stress; however, whether there exist other pathways requires further research to reveal. The overall conclusion is that there exists a pressure-dependent relationship between the increases in cellular senescence and hypertension.

In metabolic syndrome patients, hypertension is usually accompanied by insulin resistance. Under physiological conditions, insulin may regulate GFR by expanding local renal vessels. In patients with MetS, this effect will be weakened, or may even disappear. Therefore, insulin resistance can accelerate renal sclerosis in hypertension [33]. Moreover, hyperinsulinemia caused by insulin resistance can stimulate the secretion of endothelin-1, affect the function of endothelial cells, reduce the release of NO, and cause vasoconstriction. As mentioned above, cell aging promotes insulin resistance, which may indirectly accelerate the development of hypertension.

On the other hand, as an important cause of hypertension-related kidney diseases, renal artery stenosis can reduce the blood supply to the kidney and aggravate kidney damage. Studies have observed that MetS promotes microvascular remodeling, manifested as atherosclerosis of arteries and arterioles, and accelerates the development of atherosclerotic lesions [25]. MetS accelerated microvascular loss and, in turn, aggravated tissue damage in narrowed pig kidneys [26]. In MetS, oxidative stress and increased inflammation are common, which may promote the synergy between MetS and renal ischemia in this regard, with the development of hypertension-related kidney diseases.

In patients with hypertension, various approaches to regulate blood pressure often result in a dysfunctional state. When the pressure is transferred to the glomeruli, it will accelerate kidney injury, leading to glomerular sclerosis. For example, a loss of sodium regulation leads to increased sodium content in the body, while activation of RAAS also enhances sodium reabsorption. Under these conditions, NO release is reduced, which promotes the inflammatory process and induces cell senescence [27].

In their 2005 study, Imanishi et al. found that the endothelial progenitor cell (EPC) level was increased in SA-β-GAL-positive patients with hypertension [28], while telomerase activity was decreased. The mechanism by which hypertension promotes EPC aging is still unclear, but it can be speculated that high blood pressure will induce cellular senescence in EPCs. EPCs can prevent the progression of atherosclerosis by secreting growth factors, cytokines, and other vascular injury response factors. Undoubtedly, hypertension promoting the aging of EPC will intensify the progression of atherosclerosis, resulting in vascular sclerosis, which, in turn, will intensify the degree of hypertension. The glomerulus of patients with hypertension is under high pressure, such that oxidative stress and chronic inflammation are common, while RAAS activation and endothelial cell function are impaired. Additionally, it is often accompanied by indirect promotion facets, such as obesity and diabetes. Under these combined effects, the glomerular protein filtration rate is increased, and segmental necrosis and glomerular sclerosis can occur. Impairment of the glomerular filtration barrier will lead to proteinuria and podocyte reduction, which then leads to the exposure of and damage to the glomerular basement membrane [5]. With the prolongation of injury, cell stress will continue to accumulate, stimulating the process of cellular senescence. At the same time, the symptoms of kidney injury (e.g., proteinuria) will gradually worsen. High blood pressure causes kidney injury, which, in turn, aggravates high blood pressure [34]. This is the vicious cycle of hypertension-related nephropathy.

### 3.4. Hyperuricemia-Related Kidney Disease

Hyperuricemia is a common disease, mainly caused by abnormal metabolism of serum uric acid. Its pathological manifestations are mostly metabolic diseases, such as gout, hypertension, diabetes, and hypertriglyceridemia, often accompanied by metabolic syndrome. High uric acid may promote the occurrence of the above-mentioned diseases by inducing cellular senescence in various tissues. For example, the induction of endothelial cellular senescence leads to cardiovascular disease and induces the senescence of islet B cells, leading to diabetes, and so on. About two-thirds of the uric acid produced in the human body is excreted by the kidney, such that hyperuricemia is related to kidney diseases in many cases.

In observational studies, it has been shown that hyperuricemia can predict the occurrence and development of kidney disease. In animal studies, researchers have found that hyperuricemia may lead to lesions in the afferent arterioles, increased glomerular pressure, glomerular hypertrophy, and hypertension [35,36,37]. This may be related to the pro-inflammatory effects of hyperuricemia described above [38,39,40]. At the same time, hyperuricemia leads to afferent arteriopathy and tubulointerstitial fibrosis through its influence on RAAS [35].

However, the role of hyperuricemia in kidney disease is still unclear, and multiple mechanisms may be involved. The main mechanisms may be as follows: 1. The oxidation of uric acid leads to the production of reactive oxygen species (ROS) in kidney tissues, which can cause the accumulation of cellular senescence and the occurrence of lesions [41,42]; 2. uric acid blocks the release of NO, as well as inhibiting vascular expansion and the proliferation of endothelial cells [35,43,44]; and 3. uric acid can activate nuclear transcription factors, inflammatory mediators such as cyclooxygenase-2 (COX2) and tumor necrosis factor α (TNF-a) and stimulate the angiotensin–aldosterone system (RAAS) to induce smooth muscle cell proliferation, leading to endothelial cell dysfunction [42,43,44,45]. Points 2 and 3 above contribute to hypertension, which may indirectly cause hypertension-related kidney disease. In these mechanisms, many intermediate products also trigger cellular senescence, which may appear in the form of cell senescence by inducing the senescence of renal parenchyma cells and renal vascular endothelial cells. Eventually, the accumulation of senescent cells reaches a threshold, leading to pathological changes.

Uric acid is one of the most important antioxidants in the blood circulation, which can protect cells from exogenous oxidants [46,47]; however, the results of most clinical studies have shown that an elevated uric acid level is closely related to endothelial dysfunction [22,28], manifested as the decreased utilization of endothelial NO, which is a key factor in renal lesions and cardiovascular diseases [33,48]. Uric acid has a pro-inflammatory effect on endothelial cells [49,50,51,52].

The research of Yu, Min-A^a^ et al. has shown that uric acid can induce oxidative stress in endothelial cells, which can induce RASS stimulation and produce positive feedback to further enhance ROS production, thus creating a storm of oxidative stress, which will lead to the inhibition of endothelial cell proliferation, cell aging, and apoptosis.

In summary, although uric acid is considered to have an important antioxidant effect, in the hyperuricemia environment, uric acid may exert the opposite effect—that is, causing oxidative stress, blocking the release of NO, and activating RAAS, among other effects. Additionally, there may exist other unknown mechanisms, which, in turn, cause the cellular senescence of various cells. Eventually, the accumulation of cell aging exceeds a threshold, after which various pathological changes occur.

## 4. Conclusions and Future

More and more studies have shown that cellular senescence plays an important role in metabolic-associated nephropathy. Through stimulation by various cell stress factors, senescent cells gather in diseased tissues, resulting in inhibited tissue regeneration. The senescent cells secrete SASP, which affects the micro-environment and metabolic function of the surrounding tissues (e.g., causing insulin resistance), as well as causing aseptic inflammation. The excessive expansion of adipose tissue depends on a large number of replications, which contributes to the occurrence of cell senescence. Aging adipose tissue cells can affect the surrounding tissues through paracrine signaling, gradually creating a chronic inflammatory environment. In the development of diabetes and diabetic nephropathy, various cell stress factors may directly cause cellular senescence or down-regulate anti-aging proteins (e.g., Sirt1 and Klotho) and indirectly cause cellular senescence; however, the specific mechanism remains to be elucidated. The previous main argument was that cellular senescence inhibits the proliferation of islet B cells, but the results of the 2016 study by Helman et al. opened questions that require more studies to answer [23]. Age has always been a major risk factor for cardiovascular disease, and premature aging is often observed in the vascular system of hypertensive patients. Cellular senescence of EPC can promote the progression of hypertension. In the future, the risk of hypertension may be judged through the testing of the aging degree. In addition, the metabolic diseases discussed above may occur simultaneously or promote each other, where cellular senescence plays an important role in the development of these combined diseases. For example, cell senescence occurs in the adipose tissues of obese people, following which the release of SASP affects the surrounding tissues, promoting the production of an inflammatory environment and IR, which may play a role in inducing the development of diabetic nephropathy. In the development of diabetes, after the aging of islet B cells, their secretion function is reduced. Under the promotion of IR, it is difficult to control hyperglycemia, and this will finally lead to the occurrence of diabetes. A high-glucose environment will also stimulate the endothelial cells and promote their dysfunction, thus leading to the development of hypertension. Moreover, the high-uric acid environment in hyperuricemia will widely increase cell senescence, leading to the occurrence of CS in various organs and tissues (Figure 2).

As cellular senescence plays an important role in the progression of diseases, therapies targeting senescent cells naturally have great potential. Possible directions for research include the following: preventing the occurrence of cellular senescence, reversing cellular senescence, inhibiting the effect of SASP, and eliminating aging cells. Drugs aimed at preventing or reversing cellular senescence target certain regulatory factors, such as SIRT1. The anti-aging effects of metformin, pioglitazone, and SGLT2 inhibitors in diabetes have been reported, but the specific pharmacology is still unknown. In recent years, inflammation and anti-inflammation have become important aspects of the research on aging. Aging is often accompanied by low-grade chronic inflammation. Anti-inflammation may provide an effective means to fight aging. Inhibition of SASP mainly inhibits pro-inflammatory SASP, thus improving the aseptic inflammatory environment and cellular senescence. The review by Malaquin et al. has introduced some studies on drug inhibition of SASP published in recent years. All of the above research directions have certain limitations, and the most advanced method involves inducing apoptosis in senescent cells. Studies have shown that inducing senescent cell death in mouse models may be beneficial to health. In recent years, some drugs that induce the apoptosis of senescent cells have appeared [49,53,54]; however, in general, research on targeted therapies for cellular senescence to improve metabolic nephropathy is limited, and there is still great potential in this field.

## Figures and Tables

**Figure 1 cells-11-03443-f001:**
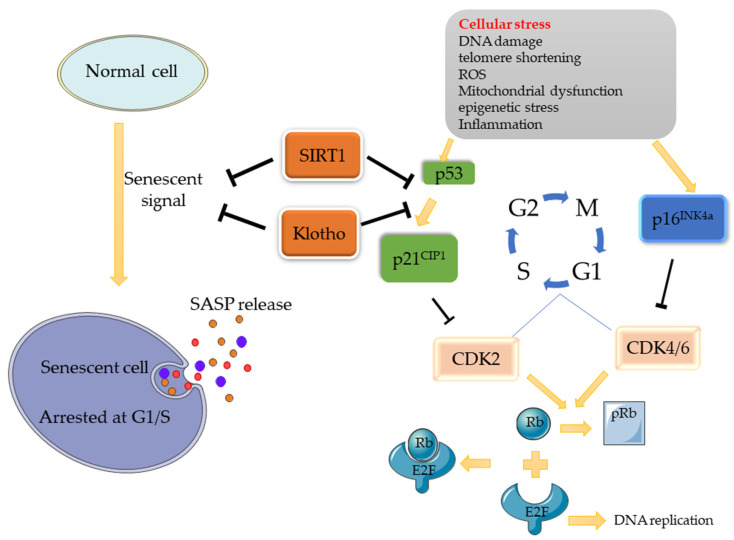
Potential Signaling Pathway of Cell Senescence. When normal cells are stimulated by cell stress, the p53/p21 and p16 pathways will be activated, thus inhibiting the activity of CDK, and, consequently, inhibiting the phosphorylation of Rb. Rb binds to E2F, which promotes DNA replication, and inactivates it, eventually causing cell replication arrested at the G1/S phase, where cellular senescence occurs. As senescence regulatory factors, SIRT1 can deacetylate p53 and interfere with its nuclear translocation process, while Klotho can also inhibit senescence signals by inhibiting the p53/p21 pathway, thereby regulating cell senescence.

**Figure 2 cells-11-03443-f002:**
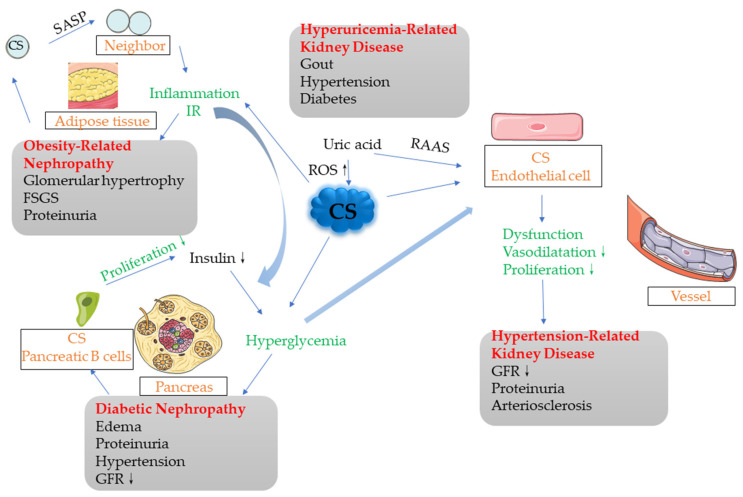
Potential associations among various metabolic-related nephropathy factors. There exist associations among multiple metabolic-related nephropathy factors, which may affect each other or simply occur simultaneously. The cells in adipose tissues generate cellular senescence and release SASP, which, in the form of paracrine signaling, affect the surrounding normal tissues and cells, gradually producing an inflammatory environment that promotes IR, leading to obesity-related nephropathy in severe cases. The middle part of this process also intersects with DN, where IR is an important factor in the development of DN. Islet B cells become old and their proliferation is blocked. If the insulin secreted by the body is not sufficient to compensate for the hyperglycemia, diabetes will occur. IR also plays a facilitating role in this process. The high-glucose environment stimulates the endothelial cells, which may promote endothelial cell aging and dysfunction, finally leading to the occurrence of hypertensive nephropathy. In addition, the high uric acid levels in the hyperuricemia environment can also lead to the generation of cellular senescence in various cells, through increasing ROS and other means, thus promoting the development of various metabolic nephropathies.

## Data Availability

Not applicable.

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
