# Peer review of "Cellular Senescence in Metabolic-Associated Kidney Disease: An Update"

_cells, 2022, doi:10.3390/cells11213443_

Round 1

Reviewer 1 Report

This review reported cellular senescence in metabolic-associated kidney disease, and the authors mainly described the relationship between cellular senescence and obesity-related nephropathy, diabetic nephropathy, hypertension- and hyperuricemia-related kidney disease, and generally speaking the effects between cellular senescence and metabolic kidney disease is mutual. This review has certain guiding significance for the research and application of anti-aging therapy in metabolic kidney disease. However, there are some problems, which must be solved before it is considered for publication.

1. Some sentences contain grammatical mistakes or are not complete sentences, such as, in line 153-155 “According to WHO standards, a BMI over 30 years old is obesity.” There is at least one Spelling error in the manuscript, such as, in line230-232 “A large number of studies have proved that cellular senescence can inhibit B cell proliferation and lead to T2DM mainly though p16INK4a pathway.” “though” would be “through”. The manuscript needs careful editing and particular attention to English grammar, spelling, and sentence structure.

2. There are a lot of descriptions about the relationship between diabetes, obesity and so on with cellular senescence, but things about kidney itself are barely little involved in this paper. Another obvious problem with this paper is lack of sufficient explanation of mechanism. As a review, possible mechanisms should be exemplified as much as possible, and the original article should be cited, too.

3. The figures in your paper are a bit blurry. Please consider replacing them with clearer ones. Or you can divide figure.2 into different parts according to different types of kidney diseases.

Author Response

Response to Reviewer 1 Comments

Point 1: Some sentences contain grammatical mistakes or are not complete sentences, such as, in line 153-155 “According to WHO standards, a BMI over 30 years old is obesity.” There is at least one Spelling error in the manuscript, such as, in line230-232 “A large number of studies have proved that cellular senescence can inhibit B cell proliferation and lead to T2DM mainly though p16INK4a pathway.” “though” would be “through”. The manuscript needs careful editing and particular attention to English grammar, spelling, and sentence structure.

Response: Thank you for your useful comments. We carefully checked and corrected the grammar and spelling mistakes and optimized the sentence structure and made the corresponding modifications on our manuscript marked up using the “Track Changes” function in MS Word after using editing service at https://www.mdpi.com/authors/english. For line 153-155 “According to WHO standards, a BMI over 30 years old is obesity”, we have modified the sentence to “According to WHO standards, obesity occurs at a BMI greater than 30” in line157-158 (The number of pages and lines we indicate here is the correct number after setting to “unmarked” in “Track Changes” mode, and the following is the same.). For line230-232 “A large number of studies have proved that cellular senescence can inhibit B cell proliferation and lead to T2DM mainly though p16INK4a pathway.” “though” would be “through”, we have modified the sentence to “A large number of studies have shown that cellular senescence mainly inhibits the pro-life ration of B cells through the p16INK4a pathway, leading to T2DM.” in line257-258.

Point 2: There are a lot of descriptions about the relationship between diabetes, obesity and so on with cellular senescence, but things about kidney itself are barely little involved in this paper. Another obvious problem with this paper is lack of sufficient explanation of mechanism. As a review, possible mechanisms should be exemplified as much as possible, and the original article should be cited, too.

Response: Thank you for your constructive comments. In the specific classification and elaboration of the section “Cellular Senescence in Metabolic-Associated Kidney Diseases”, content about the connection between this type of metabolic disease and kidney is added, mainly involving the pathological process and how to cause kidney damage and produce clinical symptoms. For line 162-178, We described the association between obesity and kidney disease. Includes data from epidemiological studies and possible ways in which obesity can cause kidney damage to develop into obesity-related kidney disease (both directly and indirectly). For line 239-243, we have summarized the possible ways of obesity-related nephropathy in kidney injury and its relationship with cell senescence again. And pointed out that more studies are needed to prove the specific mechanism. For line 274-286, we added part of the mechanism by which high glucose levels lead to kidney damage and development of DN. For line 362-367, we supplement some mechanisms of cell senescence and kidney injury caused by the disorder of regulation system in patients with hypertension. For line 376-387, we have summarized a vicious circle in which a variety of comprehensive effects in patients with hypertension lead to kidney injury, hypertension-induced nephropathy and aggravation of hypertension. However, since the specific mechanism is complex and there is no consensus at present, it is still in the research stage. We mainly discuss the parts related to cell senescence and point out this problem in the paper.

Point 3: The figures in your paper are a bit blurry. Please consider replacing them with clearer ones. Or you can divide figure.2 into different parts according to different types of kidney diseases.

Response: The picture has been modified. At page 3, line 83, we rearranged the picture structure of Figure1, adjusted the font size, and rearranged the colors according to the classification of substances to enhance the contrast, making it easier for readers to see the difference.

Regarding figure 2, the reviewers suggested that we could make it clearer, or break it into several parts. We put several parts into a picture, the purpose is to clarify that in metabolic diseases, different metabolic diseases are not completely independent, have connection with each other, or even cause and affect each other. So, we accepted the first suggestion, no split. At page 11, line 466, we enlarged the font size of figure2 by 4. There are many materials, processes and organizations in the figure. We assigned different colors according to different classifications and added the logo to make it more clear under the premise of not crowded.

Reviewer 2 Report

Cell senescence is not only a phenotype, but also an important initiator for multiple chronic diseases. In the manuscript, the authors well summarized the recent outcome in the role and underline mechanism of cell senescence in metabolic-associated kidney disease. The manuscript was well organized and provided essential information on this topic.

There are only minor concerns for the authors to address:

1. The section of signaling pathway(Figure1)could be deepened regarding the p53/p21 pathway. The authors mentioned SIRT1 is an inhibitor for cellular senescence. It could inhibit p53/p21 pathway via deacetylation of p53, which means SIRT1 is very important for cellular senescence. I would suggest adding “SIRT1” in the figure1.

2. The English language still needs to be improved before re-submission. Such as “There are two mainly pathways” should be “There are two main pathways” (page 2 line 24); “hyperglycemia is difficult to control” should be “it is difficult for ....to control hyperglycemia” (page 9 line 37)

3. Double check the usage of abbreviations, such as “GFR” in abstract need to be spelled out.

Author Response

Response to Reviewer 2 Comments

Point 1: The section of signaling pathway(Figure1)could be deepened regarding the p53/p21 pathway. The authors mentioned “SIRT1” is an inhibitor for cellular senescence. It could inhibit p53/p21 pathway via deacetylation of p53, which means SIRT1 is very important for cellular senescence. I would suggest adding “SIRT1” in the figure1.

Response: Thank you very much for your valuable comments. At page 3, line 83(The number of pages and lines we indicate here is the correct number after setting to “unmarked” in “Track Changes” mode, and the following is the same.), we have rearranged figure1 to include SIRT1 and Klotho, and added corresponding instructions in figure legend.

Point 2: The English language still needs to be improved before re-submission. Such as “There are two mainly pathways” should be “There are two main pathways” (page 2 line 24); “hyperglycemia is difficult to control” should be “it is difficult for ....to control hyperglycemia” (page 9 line 37)

Response: We carefully checked and corrected the grammar and spelling mistakes and optimized the sentence structure. For “There are two mainly pathways” should be “There are two main pathways”, we changed “mainly” to “main” at page 2, line 67. For line 459-469, “hyperglycemia is difficult to control” should be “it is difficult for ....to control hyperglycemia”, we modified the sentence to “Under the promotion of IR, it is difficult to control hyperglycemia, and this will finally lead to the occurrence of diabetes.

Point 3: Double check the usage of abbreviations, such as “GFR” in abstract need to be spelled out.

Response: We checked the abbreviations in the text and added the full names where they first appeared, such as “glomerular filtration rate (GFR)” in abstract, “Type 2 Diabetes Mellitus (T2DM)” at line165, “Acute kidney injury (AKI) and End-Stage Renal Disease (ESRD)” at line 167 and so on.

Round 2

Reviewer 1 Report

The authors have responsed to my all comments. Suggested to be accepted.